# Evaluation of a Synthetic Retinoid, Ellorarxine, in the NSC-34 Cell Model of Motor Neuron Disease

**DOI:** 10.3390/ijms25189764

**Published:** 2024-09-10

**Authors:** Olivia Escudier, Yunxi Zhang, Andrew Whiting, Paul Chazot

**Affiliations:** 1Department of Biosciences, Wolfson Research Institute for Health and Wellbeing, Durham University, Durham DH1 3LE, UK; oescudier@gmail.com (O.E.); yunxi.zhang@durham.ac.uk (Y.Z.); 2Department of Chemistry, Science Laboratories, Durham University, Durham DH1 3LE, UK

**Keywords:** Ellorarxine, DC645, retinoid, Amyotrophic lateral sclerosis, neuroinflammation, neurodegeneration, neuroprotective effects

## Abstract

Amyotrophic lateral sclerosis (ALS) is the most common motor neuron disease worldwide and is characterized by progressive muscle atrophy. There are currently two approved treatments, but they only relieve symptoms briefly and do not cure the disease. The main hindrance to research is the complex cause of ALS, with its pathogenesis not yet fully elucidated. Retinoids (vitamin A derivatives) appear to be essential in neuronal cells and have been implicated in ALS pathogenesis. This study explores 4-[2-(5,5,8,8-tetramethyl-5,6,7,8-tetrahydroquinoxalin-2-yl)ethylnyl]benzoic acid (Ellorarxine, or DC645 or NVG0645), a leading synthetic retinoic acid, discussing its pharmacological mechanisms, neuroprotective properties, and relevance to ALS. The potential therapeutic effect of Ellorarxine was analyzed in vitro using the WT and SOD1G93A NSC-34 cell model of ALS at an administered concentration of 0.3–30 nM. Histological, functional, and biochemical analyses were performed. Elorarxine significantly increased MAP2 expression and neurite length, increased AMPA receptor GluA2 expression and raised intracellular Ca^2+^ baseline, increased level of excitability, and reduced Ca^2+^ spike during depolarization in neurites. Ellorarxine also displayed both antioxidant and anti-inflammatory effects. Overall, these results suggest Ellorarxine shows relevance and promise as a novel therapeutic strategy for treatment of ALS.

## 1. Introduction

Amyotrophic lateral sclerosis (ALS) is a fatal type of motor neuron disease (MND). ALS is a multifaceted chronic neurodegenerative disease associated with rapid progressive degeneration of upper and lower motor neurons, resulting in weakness and atrophy of muscle in the limbs, trunk and bulbar region. Up to 50% of patients with ALS also develop a degree of cognitive impairment. Hence, while ALS was historically considered a disorder affecting the motor system only, it is now recognized that non-motor features are also present. Some clinical and pathological features of ALS are shared with frontotemporal dementia (FTD), which involves impaired judgement and executive skills. This relationship has been confirmed by genetic studies, such that the two conditions are now considered to be at opposite ends of the same disease continuum [1].

Understanding the etiology and pathophysiology of ALS is critical to developing methods to prevent or mitigate the disease. Developing ways to prevent or mitigate the disease is critical. The cause of ALS is complex and includes both environmental and genetic factors. Most cases are sporadic (sALS) and only 10% are autosomal dominant, known as familial ALS (fALS). The clinical manifestations of fALS and sALS are similar and, except for family history, cannot be distinguished based on clinical history or physical signs alone [1]. As with most neurodegenerative diseases, age is the main risk factor for ALS; incidence is low under age 55, but increases substantially with age, peaking at age 75 [2].

ALS has clear motor symptoms and can be diagnosed clinically, but a large proportion of neurons die before symptoms become apparent. Because neurons are difficult to regenerate, identifying the biochemical mechanisms that precede cell death is critical to developing effective neuroprotective therapies [3]. The causative process of ALS is multifactorial and is not yet fully defined. Possible mechanisms include, but are not limited to, genetic factors, oxidative stress, excitotoxicity, neuroinflammation, and protein aggregation, as well as alterations in RNA processing and mitochondrial function [4].

Vitamin A (retinol) and its derivatives (retinoids) play important roles in a variety of important cellular functions including development, differentiation, and cell death. Retinoids can enter cells as RA or retinol molecules. Once inside the cell, RA binds to another RBP family, the cellular retinoic acid-binding proteins (CRABPs), which are responsible for ATRA catabolism (CRABP-I) and nuclear import of RA (CRABP-II). CRABP-II promotes the nuclear translocation of ATRA, allowing it to bind to retinoid nuclear receptors (NRs). There are two main families of these RA receptors: Retinoic Acid Receptors (RARs), with three isotypes (RARα, RARβ, and RARγ), and the Retinoid X Receptors (RXRs), also with three isotypes (RXRα, RXRβ, and RXRγ) [5].

In the case of ALS, retinol-regulated neuronal genes may have effects on other important cellular processes in addition to inducing neural differentiation, motor axon growth, and neural patterning. Specific examples may include antioxidant responses (SOD1), neuroinflammation and immunomodulation, intracellular signaling, and synaptic homeostasis [5]. The investigation into the association between serum retinol-binding protein 4 (RBP4) concentration as a surrogate marker for serum retinol and the risk and prognosis of ALS revealed a correlation between serum RBP4 concentration and survival [6]. This suggests a potential association between retinol levels and better prognosis in ALS patients. Furthermore, studies on retinoid agonists have demonstrated their protective potential in ALS mouse models [5]. Activation of retinoic acid receptor signaling (RARs and RXRs) has been found to play a significant neuroprotective role in ALS. All-*trans*-retinoic acid (ATRA) has been found to fully reverse impaired RARα signaling in the spinal cord of ALS mice [7]. Pretreatment of cultures enriched with primary motor neurons with either pan-RAR or RARβ-specific agonists mitigates motor neuron cell death linked to oxidative damage or stress, whereas RARβ-specific antagonists exacerbate cell death. The pro-survival and regenerative responses mediated by RARβ demonstrate protective effects against oxidative damage in neurons [8].

In this study, we aimed to evaluate the efficacy of a synthetic retinoid, Nevrargenics’ lead drug, Ellorarxine (also known as DC645 and NVG0645), an RAR-modulator, on ALS in vitro. Ellorarxine, 4-[2-(5,5,8,8-tetramethyl-5,6,7,8-tetrahydroquinoxalin-2-yl)ethylnyl] benzoic acid, was designed as a mimic of all-trans-retinoic acid in its lowest energy conformation, and therefore as a potentially potent RAR-agonist. Its design is based upon an earlier compound, EC23 [9], but with two nitrogen atoms added to the tetramethyltetrahrdonaphthalene section of the molecule, with the intention of making the resulting compound more water soluble and potent.

NSC-34 cells and NSC-34 SOD1G93A were used as model motor neuron (MN) and MND, for in vitro experiments, respectively. NSC-34 is a hybrid cell line derived from the fusion of MN-rich embryonic mouse spinal cord cells with mouse neuroblastoma. The culture contained two cell populations: small undifferentiated cells and larger multinucleated cells. Removal of serum allows NSC-34 cells to differentiate and form neurons.

## 2. Results

### 2.1. Immature and Mature NSC-34 Cells: Changes with Differentiation

To induce differentiation, illustrated by the development of long neuronal processes visible by phase-contrast microscopy, NSC-34 cells were subject to serum-free media (SFM) (Figure 1). Morphologically, cells cultured in complete media appeared small and circular immediately after passaging and before adhesion to the flask had occurred. After six-day culture in SFM, two morphologically distinct populations of NSC-34 cells were identified: cells with short neurites (Figure 1A) and cells with long processes, characteristic of the MN phenotype (Figure 1B).

### 2.2. Ellorarxine Promoted Differentiation (MAP2 Expression) and Increases Neurite Length

It is understood that microtubule-associated protein 2 (MAP2) is a protein involved in microtubule assembly, which is a key step in neurogenesis and a complex process in the formation and development of neurons [10]. Therefore, this protein is suitable for use as a marker of neuronal differentiation. Immunofluorescence revealed that the abundance of MAP2 was increased in NSC-34 cells cultured with Ellorarxine (10 nM) for 6 days compared with untreated cells, indicating that Ellorarxine successfully induced the expression of MAP2 (Figure 2A). We further investigated the ability of Ellorarxine to promote differentiation into MN-type cells and generate longer neurite processes. The results showed that the addition of Ellorarxine (10 nM) significantly increased neurite length by approximately 94%, from an average of 40 µm to 78 µm (Figure 2B).

### 2.3. Ellorarxine Pretreatment Modulated Inflammation and Inflammatory Cytokine Release

To assess the ability of Ellorarxine to modulate inflammation, we analyzed the microglial tethering marker CD200. Immunofluorescence assays showed higher expression levels of CD200 in Ellorarxine-treated cells, suggesting that Ellorarxine prevents microglial activation by altering the expression of this tethering marker (Figure 3).

To investigate the further effect of Ellorarxine on neuroinflammation, a cytokine release assay was performed. Lipopolysaccharide (LPS) was used to induce inflammatory stress. After incubation with Ellorarxine (10 nM), a significant reduction in inflammatory signals was observed under LPS stress compared with the control. From our four replications, TNF-α release was reduced from an average of 700 pg/mL to 500 pg/mL in wildtype NSC-34 cells, while an average decrease from 800 pg/mL to 400 pg/mL was observed in mutant SOD-1 NSC-34 cells. We also observed a significant decrease in IL-6 release. IL-6 levels decreased from 600 pg/mL to 360 pg/mL in wildtype and decreased from 700 pg/mL to 300 pg/mL in mutant group (Figure 4).

The effect of Ellorarxine (10 nM) in differentiated cells was studied. In wildtype, application of Ellorarxine significantly reduced pro-inflammatory marker TNF-α release. TNF-α release was decreased from 75 pg/mL to 25 pg/mL under 0.30 μg/mL LPS stress and from 450 pg/mL to 250 pg/mL under 1.00 μg/mL LPS stress (Figure 5).

### 2.4. Ellorarxine (DC645) Treatment Rescued NSC-34 Motor Neurons from Oxidative Stress

Differentiated wild-type and mutant SOD1 NSC-34 cells showed a dose-dependent decrease in cellular mitochondrial viability after peroxide-induced stress. The IC_50_ of mutant SOD1 cells was approximately 4-fold lower than that of wildtype, 2.26 µM and 9.55 µM, respectively, indicating increased sensitivity of mutant SOD1 NSC-34 cells to oxidative stress. Therefore, the potential of Ellorarxine to protect against peroxide-induced oxidative stress was evaluated by preincubation with Ellorarxine (10 nM) for 1 h. The IC_50_ values increased for both wildtype and mutant forms, from 9.55 µM to 13.99 µM and from 2.26 µM to 6.43 µM, respectively (Figure 6).

### 2.5. Ellorarxine Treatment Increases Expression and Redistributes GluA2 in Differentiated Mutant SOD1 NSC-34 Cells

Immunofluorescence showed that the AMPA receptor subunit that controls calcium permeability and receptor mobility, GluA2, was mainly membrane localized around the cell body in wildtype and mutant controls. Administration of Ellorarxine to mutant SOD1 cells resulted in a significant increase in GluA2 expression and alterations in its topology. GluA2 staining in Ellorarxine-treated cells extended into the cellular processes. The staining pattern in treated cells also suggests that GluA2 moves radially from the nuclear periphery toward the outer membrane and into the neurites (Figure 7).

### 2.6. Ellorarxine Treatment Raised Intracellular Ca^2+^ Baseline, Increased Level of Excitability Longterm, and Reduced Ca^2+^ Spike on Depolarization in Neurites

Live single-cell Ca^2+^ imaging was used to investigate the effect of Ellorarxine on changes in baseline intracellular calcium and following KCl mediated depolarization (excitability). Imaging was performed for a total of 200 cycles, with a depolarization buffer applied after 100 cycles. Ca^2+^ imaging revealed a significant difference in fluorescence intensity profile between control cells and those treated with Ellorarxine. Key changes to the profile with Ellorarxine treatment include a raised intracellular Ca^2+^ baseline for both cytoplasmic and nuclear data, an overall increased level of excitability, and reduced Ca^2+^ spike on depolarization in neurites (Figure 8).

## 3. Discussion

This study aimed to evaluate the ability of Ellorarxine to ameliorate ALS-related pathophysiological mechanisms without adverse effects. The results confirmed that Ellorarxine can induce differentiation and neurite growth, increase mitochondrial viability to combat oxidative stress, ameliorate neuroinflammation, and regulate intracellular Ca^2+^ via modification of AMPA receptor GluA2 expression and topology.

Vitamin A and its derivatives play a vital role in neural development. Retinoic acid is essential for the proliferation of stem and progenitor cells in the adult brain, and studies have shown that it is involved in promoting cell proliferation, inducing stem cell differentiation, and regulating progenitor cell behavior [11]. Neuronal differentiation is characterized by morphological progression (i.e., neuronal growth) and functional changes (i.e., synapse formation). There is an important link between the development process and the regeneration of neuronal tissue caused by trauma to the central nervous system and neurodegenerative diseases. The differentiation process relies on the expression of different proteins, such as neurofilament proteins and microtubule-associated proteins [12]. In motor neuronal cultures treated with Ellorarxine, MAP2 expression increased (Figure 2), which is consistent with previous reports that retinoic acid mediates the generation of specific neuronal cell types and the maintenance of a differentiated state [13]. This suggests that Ellorarxine has similar effects on inducing cell differentiation as other retinoids. Multiple lines of evidence from human and animal studies have demonstrated that ALS is associated with both inflammatory (local, innate immunity) and immune responses (peripheral adaptive responses), with tumor necrosis factor-α (TNF-α), TNF receptor 1 (TNFR1), interleukin 6 (IL-6), IL-1β, IL-8, and vascular endothelial growth factor (VEGF) measured in the blood significantly elevated in ALS cases compared with controls. Studies have found that IL-6 has one of the highest diagnostic accuracies among the biomarkers studied in distinguishing ALS patients from healthy controls [14]. Therefore, IL-6 has also been studied as a therapeutic target for ALS, and Phase II clinical trials of IL-6 blockers are already underway. Previously, RA and synthetic retinoid Am80 have been shown to inhibit IL-6 expression and signal transduction [15,16]. Similarly, our results showed that Ellorarxine can significantly reduce LPS-induced IL-6 expression in Wild type SOD1 and Mutant SOD1 NSC-34 cells, which is not only consistent with previous reports of other synthetic retinoids, but also suggests that Ellorarxine may be able to successfully reduce the chronic inflammatory response observed in ALS.

Research results have shown that the reduction of anti-inflammatory CD200/CD200R is a potential mechanism leading to chronic inflammation in neurodegenerative diseases, including ALS [17]. CD200-CD200R1 inhibitory mechanisms remain a relatively unexplored field of science. Studies in animal models have shown a decrease in CD200 expression in neurodegenerative diseases, supporting it as a therapeutic target [18,19]. In contrast, human data are limited, with a suggestion that CD200R1 expression is more significant, implying the interaction partner may be of greater importance. Moreover, Appel et al. recently identified a role for mRNA splicing in the regulation of CD200-CD200R1 complexation, highlighting that further complexity exists within the system [20]. From our immunofluorescence image results, Ellorarxine has the potential to ameliorate some chronic neuroinflammation mediated by microglial activation by maintaining the CD200-CD200R1 inhibitory mechanism. This has rarely been reported in other synthetic retinoids and requires further study.

Although the mechanism of motor neuron degeneration is multifactorial and complex, there is growing evidence to support the hypothesis that oxidative stress is one of the mechanisms underlying motor neuron death. Antioxidants have been continuously tested as a potential treatment for ALS [21]. Studies have long shown that retinoic acid can reduce oxidative stress and cell apoptosis by maintaining the levels of SOD-1 and SOD-2 proteins, thereby showing an antioxidant effect [22]. Hydrogen peroxide, a membrane-permeable oxidant, can induce neuronal injury as it is a precursor to the production of ROS, including hydroxyl radicals. Thus, H_2_O_2_ was employed as an oxidative toxicity stimulus to examine the protective role for Ellorarxine, i.e., investigating the potential of Ellorarxine to protect against H_2_O_2_-induced oxidative stress. It was found that the IC_50_ values of both wild-type and mutant types increased after preincubation with Ellorarxine (3–5 fold), indicating that Ellorarxine successfully reduced the sensitivity of both cells to this stressor. Retinoic acids are known to have antioxidant properties, and the results demonstrate the same capabilities of Ellorarxine [23].

Glutamate-induced AMPA receptor (AMPAR)-mediated excitotoxicity is one of the causes of selective degeneration of MN in ALS [24]. It is known that MNs are particularly susceptible to Ca^2+^ influx through AMPA receptors, whose Ca^2+^ permeability is determined by the GluA2 subunit of the receptor complex. Our immunofluorescence results showed that after Ellorarxine treatment, the expression of GluA2 was significantly increased and migrated to the outer cell membrane. In the late stage of synaptic plasticity, upregulated GluA2 expression and translocation of GluA2 to synapses is known to occur, which is consistent with our initial observations [25]. This suggests that Ellorarxine may act as an inducer of homeostatic synaptic plasticity [26].

In neurons, Ca^2+^ plays the dual role of charge carrier and intracellular messenger. The diversity of Ca^2+^ signaling mechanisms in terms of amplitude and spatiotemporal pattern enables a ubiquitous messenger to regulate a large number of developmental processes, neurotransmitter release, and membrane excitability [27]. Although the mechanisms leading to selective degradation of MN in ALS are unclear, pathogenic features including glutamate-mediated excitotoxicity, formation of Ca^2+^-dependent cytoplasmic protein aggregates, and Ca^2+^-induced mitochondrial dysfunction are thought to play a key role [28]. ALS spinal cord MNs do not express the Ca^2+^-binding proteins, parvalbumin, and calbindin D28k, but express abundant GluA2-deficient AMPARs, resulting in excessively elevated free intracellular Ca^2+^ [29].

Three key changes in the intracellular Ca^2+^ signal profile occurred after Ellorarxine administration, including an increase in the intracellular baseline for data, an increase in overall excitability levels, and a decrease in Ca^2+^ spikes upon neuronal depolarization. In contrast, previous studies have demonstrated that RA induces a rapid decrease in intracellular Ca^2+^ concentration and reduces current through VGCCs [30]. However, it should be noted that due to the rigidity of Ellorarxine itself, this compound may bind to a subset of the RARs (e.g., RARß), and therefore elicit different effects. This idea is supported by the fact that RA is able to alter the firing potential of neurons, whereas ATRA or 9-cis RA are not [30]. Observations from live cell imaging suggest that Ellorarxine increases baseline intracellular Ca^2+^ levels and induces a raised excitable state. This may have positive implications regarding increased synaptic plasticity (LTP). This study has limitations. The mechanisms underpinning the positive effects of Ellorarxine on NSC-34 cells remains to be determined, and its selective activation on RAR subtypes also needs verification (RARß). Simultaneously, most synthetic retinoids have side effects on the skin and other organs, requiring the toxicity of Ellorarxine to be tested in other cell models. Progress in investigating such effects is underway and will be explored in subsequent papers.

In summary, we have provided new evidence that the RAR-M, Ellorarxine, displays a wide-range of positive effects in a commonly used in vitro SOD1G93A NSC-34 mouse cell model of ALS. This addresses the three Ns: neuroprotection, neuroplasticity, and neurorepair. The MHRA provided scientific advice for the design of the First in Human study and the non-clinical plan for Ellorarxine for ALS/FTD.

## 4. Materials and Methods

### 4.1. Cell Lines and Culture

Mouse Motor Neuron (NSC-34) were obtained from Durham University and cultured in Dulbecco’s modified Eagle’s medium (DMEM, Gibco, London, UK) supplemented with 10% fetal bovine serum (FBS, Gibco) and 1% Penicillin Streptomycin Solution (Pen-Strep, Lonza, Bend, OR, USA) at 37 °C in a humidified 5% CO_2_ incubator. The growth medium was changed every 2 days. When the culture reached 80% confluence, trypsin-EDTA was added and incubated for 2 min to make adherent cells detached. Fresh media (6 mL) was added to the incubated cells, aspirated into a flacon tube and centrifuged at 1000 rpm for 5 min at room temperature. Supernatant was discarded, and the cellular pellet re-suspended in fresh growth media. Cells were then seeded 1:3 or 1:4 into 24-well plates or T75 flasks for further growth.

After 24 h of culture, medium were replaced with FBS-free DMEM/F-12 medium. This medium is also called serum-free medium (SFM). This medium is added to induce morphological differentiation. The cells need to be cultured for 6 days as the optimal time limit.

### 4.2. Preparation of Ellorarxine

Ellorarxine (1 mM in DMSO) was obtained from Nevrargenics and was stored at −20 °C. The drug was prepared to 1 μM stock solution using dH_2_O and was stored at 4 °C.

### 4.3. Pre-Treatments

After trypsinization, cells were plated (40,000 cells/mL) in 24-well plate chambers and left to grow for 24 h at 37 °C and 5% CO_2_ before being treated with 10% DMSO (Merck Life Science UK Limited, Gillingham, UK) or 10 nM Ellorarxine for 1 h before being stressed.

### 4.4. Methyl Thiazolyl-Diphenyl-Tetrazolium Bromide (MTT) Assay

To begin, 100 μL of 2.5 mg/mL MTT (298-93-1, Merck Life Science UK Limited, Gillingham, UK) was added to each well and left to incubate for 4 h at 37 °C and 5% CO_2_. Following incubation, the contents of each well were transferred to individual Eppendorf tubes and subject to centrifugation at 3000 rpm for 10 min. MTT-containing supernatant was aspirated, and the pellet re-suspended in 250 μL isopropanol. Finally, 100 μL from each Eppendorf tube was transferred to a 96-well tissue culture plate and the absorbance was measured at 595 nm, using a microplate reader.

### 4.5. Lactate Dehydrogenase (LDH) Release Assay

LDH release was measured using CytoTox 96 kit (ADG1781, Promega, Southampton, UK). Samples (100 μL) were collected from each well before and after 72 h of freezing. All samples, collected both prior to and post freezing, were subjected to centrifugation at 1000 rpm for 2 min, diluted in dH_2_O (1 in 3 dilution) and then plated in duplicate into a 96-well plate for the assay. Then, 50 μL samples and 50 μL substrate mix were added, and the reaction was stopped by adding 50 μL of stop solution. Subsequently, the optical density was measured at 490 nm.

### 4.6. Enzyme-Linked Immunosorbent Assay (ELISA)

Then, 24 h after stressing the cells, 100 μL of the supernatant was collected from each well and ELISA was carried out using the Human IL-6 ELISA kit (ab222503, Abcam, Cambridge, UK) and Human TNF-α ELISA kit (ab208348, Abcam) according to the manufacturer’s protocol. The standard curve generated was used to calculate concentrations from the absorbance measurements.

### 4.7. Live Single Cell Ca^2+^ Imaging

Since the protocol requires photosensitive elements, all imaging preparations were performed in the dark and covered with aluminum foil. Additionally, to minimize cell shaking, all solutions used at any stage of the protocol were maintained at a temperature of 37 °C. Cells were plated into μ dishes 8000/mL with serum removed and cultured for 6 days. Incubate with Ellorarxine (10 nM) for 24 h before imaging. Afterwards, remove the medium and wash the cells with 1× HEPES physiological buffer (150 mM NaCl, 1 mM MgCl_2_, 10 mM HEPES, 2 mM CaCl_2_, 5 mM KCl). Mix Ca^2+^ Green probe (6 µL, 2 mM, Fisher Scientific, Loughborough, UK) in DMSO (19.4 µL) with 1× HEPES physiological buffer (6 mL). Ca^2+^ Green probe solution (1 mL) was added to each microwell, and cells were incubated for 40 min. During incubation, KCl depolarization buffer (0.369 g KCl in 10 mL HEPES physiological buffer) was heated in a water bath at 37 °C, and imaging was performed using a Zeiss 880 equipped with an Airyscan confocal microscope and software. UV excitation of the Ca^2+^ probe produced a signal that was recorded for 200 cycles per µ disk at 1 s intervals. KCl depolarization buffer (111 µL) was added after 100 cycles. Analyses were performed using Zen 2 (Blue Edition) imaging software (Jena, Germany).

### 4.8. Immunofluorescence STAINING

Cells were plated 8000/mL in 6-well (35 mm) chambers onto 15 mm × 15 mm coverslips. Then, 24 h after treatment, the cells were fixed in 4% paraformaldehyde (PFA) for 10 min at room temperature. Cells were washed three times for 5 min with PBS and then blocked in PBS containing 1% bovine serum albumin, 1% fish skin gelatin, and 0.3% Triton X-100 at room temperature for 1 h. Then the cells were incubated with primary antibody for 1 h at room temperature. Primary antibodies were diluted as: Microtubule-associated protein 2 (MAP2) (1:200, Cell Signaling Technology, 44542, Leiden, The Netherlands), Glutamate Ionotropic Receptor AMPA Type Subunit 2 (GluA2) (1:200, Cell Signaling Technology, 5306), Cluster of differentiation 200 (CD200) (1:1000, Abcam, ab333734). Cells were then washed three times for 5 min in PBS and incubated with secondary antibodies (goat anti-rabbit Alexa Fluor^®^ 594, 1:1000, Thermo Fisher, A32740, Loughborough, UK) for 1 h at room temperature. Cells were then washed three times for 5 min with PBS and incubated with DAPI (1 μg/mL) for 5 min at room temperature to stain the DNA for nuclear localization. Fluorescent images were captured by using a Zeiss fluorescent microscope (Zeiss ApoTome 3, Cambridge, UK).

### 4.9. Statistical Analysis

Data were processed with Microsoft Excel 2019 (Microsoft, Redmond, WA, USA), and all statistical analyses were performed using GraphPad Prism software version 8.1 (GraphPad Software, La Jolla, CA, USA). All quantitative data are expressed as mean ± standard deviation of the mean. Data were subjected to one-way analysis of variance (ANOVA) and post hoc tests were performed. IF data were analyzed using Image J, 1.54J software (Fiji Computer Software, National Institutes of Health, Bethesda, MD, USA). Add graticules to evenly adjust the fluorescence channels used and the brightness and contrast of the composite image. Measure against the graticule using Image J.

## Figures and Tables

**Figure 1 ijms-25-09764-f001:**
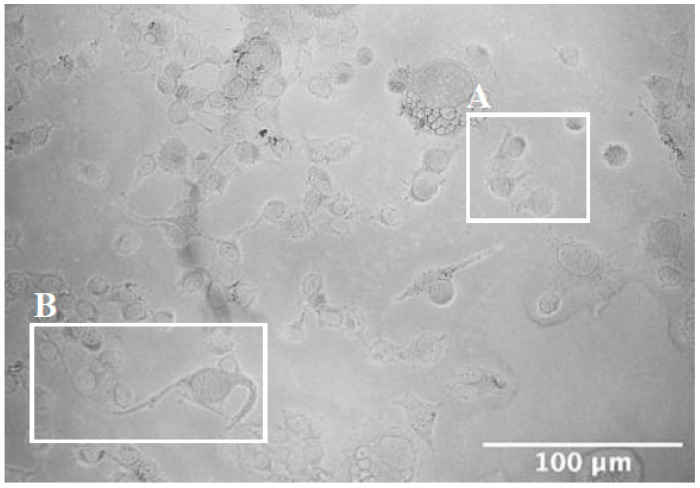
Serum deprivation reveals the presence of two morphologically distinct populations of WT NSC-34. Phase contrast microscopy images revealed two populations of NSC-34 present in Serum-Free Media (SFM): cells with short neurites (**A**) and those with longer branching processes (**B**). Imaging was carried out at one magnification, ×20, on the Zeiss Apotome microscope. Scale bars = 100 μm.

**Figure 2 ijms-25-09764-f002:**
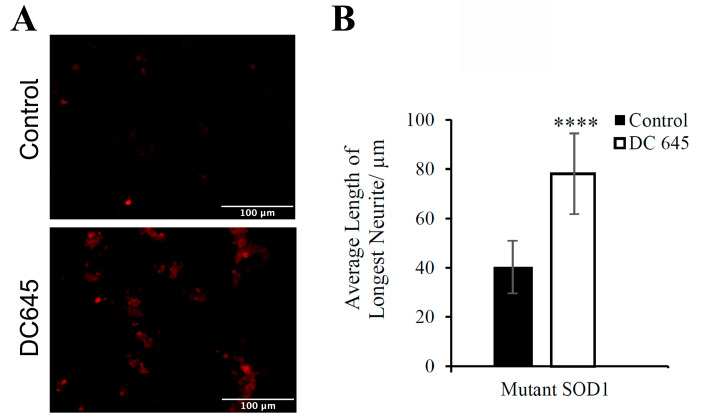
(**A**) Effects of Ellorarxine (DC645) on MAP2 semi-quantitative immunofluorescent staining of differentiated WT NSC-34 cells, (**B**) Morphological differentiation of undifferentiated NSC-34 cells determined by quantitative measurement of neurite length (α-MEM). (**A**) Immunofluorescence revealed MAP2 expression was more prevalent in differentiated cells treated with Ellorarxine than without (representative figure from n = 4 replicates). Imaging was carried out at one magnification, ×20, on the Zeiss Apotome microscope. Cells without any primary antibody were used as a negative control. Scale bars = 100 µm. (**B**) A significant increase is observed for mutant SOD1 NSC-34 cells cultured with Ellorarxine (10 nM). Data expressed as means ± SD. (n = 10) (**** *p* < 0.0001).

**Figure 3 ijms-25-09764-f003:**
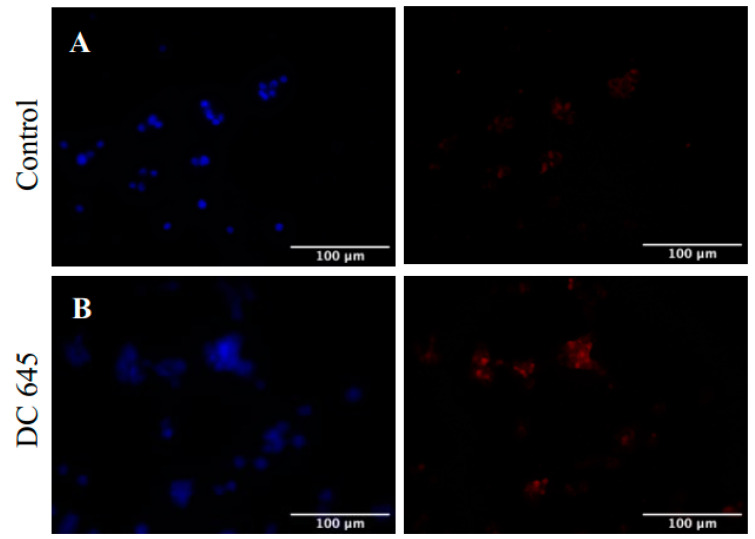
Effects of Ellorarxine (DC645) on CD200 immunofluorescent staining of differentiated WT NSC-34 cells. Staining of both untreated (**A**), control, cells, and those treated with Ellorarxine (10 nM) (**B**) for DAPI (left, blue) and CD200 (right, red). Semi-quantitative immunofluorescence revealed CD200 expression was more prevalent in differentiated cells treated with Ellorarxine than without (representative figure from n = 4 replicates). Imaging was carried out at one magnification, ×20, on the Zeiss Apotome microscope. Cells without any primary antibody were used as a negative control. Scale bars = 100 µm.

**Figure 4 ijms-25-09764-f004:**
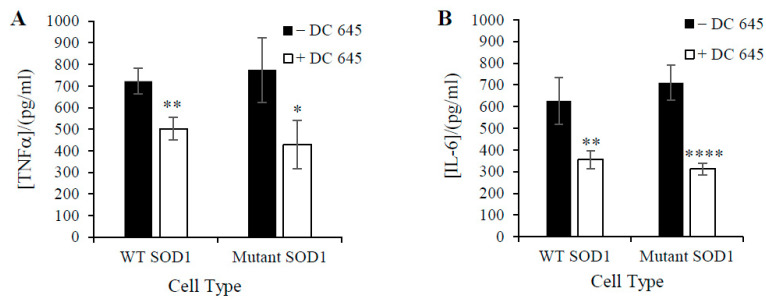
Effects of Ellorarxine (DC645) on TNF-α (**A**) and IL-6 (**B**) release of undifferentiated NSC-34 cells incubated with 0.3 µg/mL LPS. NSC-34 cells were pre-treated with 10 nM Ellorarxine for 1 h before 24 h of inflammatory insult with 0.3 µg/mL LPS. Data expressed as means ± SD. (n = 4) (* *p* < 0.05, ** *p* < 0.01, **** *p* < 0.0001).

**Figure 5 ijms-25-09764-f005:**
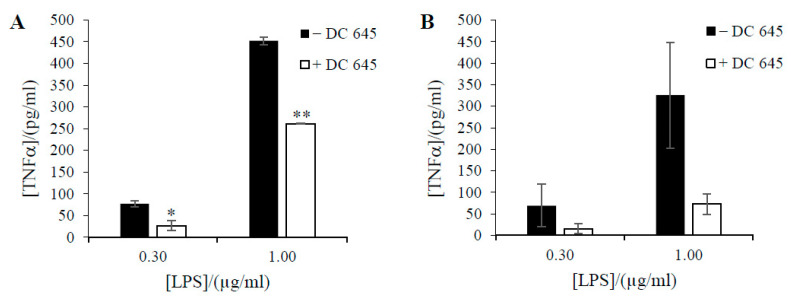
Effects of Ellorarxine (DC645) on TNF-α release of differentiated NSC-34 cells incubated with LPS. Release of pro-inflammatory signal in differentiated wild type (**A**) and mutant (**B**) NSC-34 cells with and without a 1-h pre-incubation with Ellorarxine. Data expressed as means ± SD. (n = 4) (* *p* < 0.05, ** *p* < 0.01).

**Figure 6 ijms-25-09764-f006:**
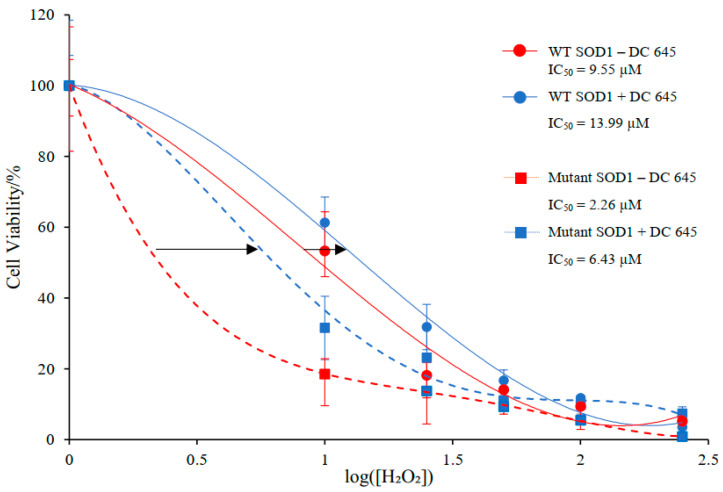
Effects of Ellorarxine (DC645) on the dose response effect of H_2_O_2_ on differentiated wild type and mutant SOD1 NSC-34 cells. Cell viability, compared to control, for cells treated with H_2_O_2_ for a 24-h incubation (red) and after a 1-h pre-incubation with Ellorarxine (blue). Arrows indicate the change in IC_50_ when treated with Ellorarxine.

**Figure 7 ijms-25-09764-f007:**
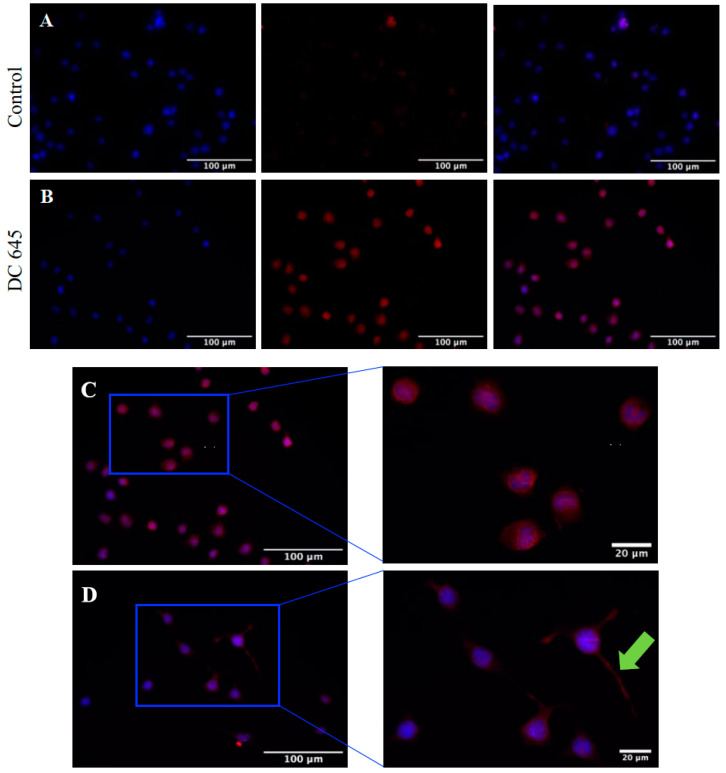
Effects of Ellorarxine (DC645) on GluA_2_ immunofluorescent staining of undifferentiated and differentiated mutant SOD1 NSC-34 cells. Staining of both untreated (**A**,**C**), control, cells, and those treated with Ellorarxine (10 nM) (**B**,**D**) for DAPI (left, blue) and GluA2 (centre, red) as well as merged channel images (right). Immunofluorescence revealed GluA2 expression was more prevalent in differentiated cells treated with Ellorarxine than not, with clear expression neurite outgrowths (green arrow). Imaging was carried out at one magnification, ×20, on the Zeiss Apotome microscope. Cells without any primary antibody were used as a negative control. Scale bars = 100 µm.

**Figure 8 ijms-25-09764-f008:**
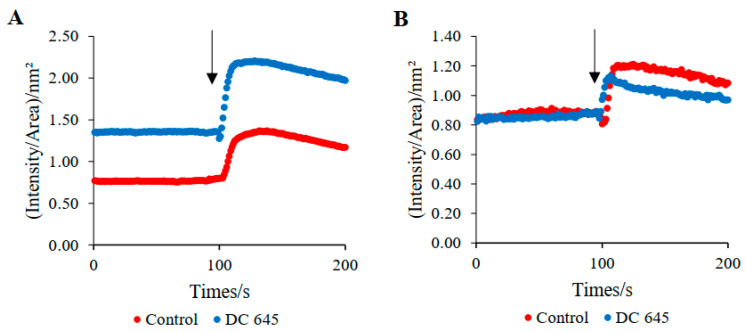
Ca^2+^ Imaging of differentiated NSC-34 cells: Effects of Ellorarxine (DC645). Changes in normalized fluorescence intensity as a function of time for the average of 10 individual cell body (**A**), neurite (**B**). Addition of the depolarizing buffer (KCl) is indicated by an arrow (n = 10).

## Data Availability

The data presented in this study are available on request from the corresponding author.

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
