# Peer review of "Evaluation of a Synthetic Retinoid, Ellorarxine, in the NSC-34 Cell Model of Motor Neuron Disease"

_ijms, 2024, doi:10.3390/ijms25189764_

Round 1

Reviewer 1 Report

Comments and Suggestions for Authors

The text delves into amyotrophic lateral sclerosis (ALS), the most prevalent motor neuron disease worldwide, known for its progressive muscle atrophy. Despite the availability of two approved treatments, they offer only temporary relief from symptoms and do not provide a cure. The intricate nature of ALS's causes and our incomplete understanding of its development present significant challenges to advancing research. The highlighted study in the text investigates the potential therapeutic effects of a synthetic retinoic acid derivative, Ellorarxine (DC645), on ALS. The study utilized the SOD1G93A NSC-34 cell model of ALS to explore Ellorarxine's pharmacological mechanisms and neuroprotective properties.

The findings suggest that Ellorarxine shows potential as a treatment for ALS by improving neuronal structure, modulating receptor expression and calcium dynamics, and providing antioxidant and anti-inflammatory effects.

Strengths and limitations:

- The use of the well-established SOD1G93A NSC-34 cell model for studying ALS enhances the findings' credibility.

- The comprehensive evaluation, encompassing histological, functional, and biochemical assessments, strengthens the understanding of Ellorarxine's effects.

-  This study is performed in an artificial environment outside of a living organism, and the findings may not directly apply to real-life conditions or human patients.

Conclusion

The study suggests that Ellorarxine holds promise as a potential therapy for ALS. It demonstrates multiple neuroprotective effects. However, more research, including in vivo studies and clinical trials, is needed to confirm these findings and to explore the potential therapeutic benefits of Ellorarxine for ALS patients.

Please review the text in lines 59-62 as there is a repetition and also the sense of the text is not very clear:

There are two main families of these receptors, each with three different subtypes. There are two main families of these receptors, each with three different subtypes: RA receptors (RARα, RARβ, and RARγ) and retinoids. Retinoid X receptors (RXRα, RXRβ, and RXRγ).

Author Response

The text delves into amyotrophic lateral sclerosis (ALS), the most prevalent motor neuron disease worldwide, known for its progressive muscle atrophy. Despite the availability of two approved treatments, they offer only temporary relief from symptoms and do not provide a cure. The intricate nature of ALS's causes and our incomplete understanding of its development present significant challenges to advancing research. The highlighted study in the text investigates the potential therapeutic effects of a synthetic retinoic acid derivative, Ellorarxine (DC645), on ALS. The study utilized the SOD1G93A NSC-34 cell model of ALS to explore Ellorarxine's pharmacological mechanisms and neuroprotective properties.

The findings suggest that Ellorarxine shows potential as a treatment for ALS by improving neuronal structure, modulating receptor expression and calcium dynamics, and providing antioxidant and anti-inflammatory effects.

Strengths and limitations:

- The use of the well-established SOD1G93A NSC-34 cell model for studying ALS enhances the findings' credibility.

- The comprehensive evaluation, encompassing histological, functional, and biochemical assessments, strengthens the understanding of Ellorarxine's effects.

-  This study is performed in an artificial environment outside of a living organism, and the findings may not directly apply to real-life conditions or human patients.

Conclusion

The study suggests that Ellorarxine holds promise as a potential therapy for ALS. It demonstrates multiple neuroprotective effects. However, more research, including in vivo studies and clinical trials, is needed to confirm these findings and to explore the potential therapeutic benefits of Ellorarxine for ALS patients.

We agree with reviewer. Some in vivo studies in SOD1G93A mice has been performed, and in fact this drug has been MHRA approved for use in ALS/FTD based on the data herein and our unpublished in vivo and safety data. Stated in text

Please review the text in lines 59-62 as there is a repetition and also the sense of the text is not very clear:

There are two main families of these receptors, each with three different subtypes. There are two main families of these receptors, each with three different subtypes: RA receptors (RARα, RARβ, and RARγ) and retinoids. Retinoid X receptors (RXRα, RXRβ, and RXRγ).

This has been clarified

Reviewer 2 Report

Comments and Suggestions for Authors

The article entitled "Evaluation of a synthetic retinoid, Ellorarxine (DC645), in the SOD1G93A NSC-34 cell model of ALS" by Escudier O et al., with number ijms-3100888  explores a lead synthetic retinoic acid relevant to ALS, and neuroprotective properties in vitro using the SOD1G93A NSC-34 cell model of ALS interestingly compound displayed pharmacological properties as increased MAP2 expression, increased AMPA receptor GluA2 expression and raised intracellular Ca2+ baseline, as well as antioxidant properties. Although the article is quite interesting, some major concerns should be addressed to publish the article: 

There are a few miscues on redaction and orthography, such as spaces, dots and misswords. Please correct them. 

Title 

Please avoid numerical numbers on the title; generally, text from titles should be easy to follow. Please correct. 

 Introduction

Can the chemical structure of Ellorarxine be drawn? This is quite important since it gives an idea of the main changes in the structure used to change activity. 

Results

Figure images are pretty blurred and don´t distinguish the suggested results. Please try to improve the figure or add a different microphotography; otherwise, the results will be difficult to believe. Please add to the figure legend the number of repetitions you performed in the experiments. 

Why is DAPI in Figures so blurred and morphology so changed in cells? 

Did you measure the number of cells that successfully induced the expression to have quantitative data rather than qualitative images? If not, why did you do so?

Do you have an image that demonstrates the morphological differentiation on NSC-34 cells? 

In Figure  8B, what do you mean by Process? Please use the correct terminology for Calcium ions. 

Do you evaluate the pharmacokinetic potential of Ellorarxine? 

Discussion

Please correct the discussion, which is huge and quite blurred. What are you trying to demonstrate your results? Did your hypothesis get demonstrated? What do you obtain from your results? Why do you believe such results? 

The senescence section is difficult to believe since it is not experimentally demonstrated. The authors make hypotheses without any demonstration that are entirely speculative and not quite scientific.

If RNA metabolism is essential, why did you not evaluate it on your assays? 

What do you conclude with your experiments? Are you going to improve your compound? Evaluate such on other models? Evaluate toxicity?

Comments on the Quality of English Language

The article entitled "Evaluation of a synthetic retinoid, Ellorarxine (DC645), in the SOD1G93A NSC-34 cell model of ALS" by Escudier O et al., with number ijms-3100888  explores a lead synthetic retinoic acid relevant to ALS, and neuroprotective properties in vitro using the SOD1G93A NSC-34 cell model of ALS interestingly compound displayed pharmacological properties as increased MAP2 expression, increased AMPA receptor GluA2 expression and raised intracellular Ca2+ baseline, as well as antioxidant properties. Although the article is quite interesting, some major concerns should be addressed to publish the article: 

There are a few miscues on redaction and orthography, such as spaces, dots and misswords. Please correct them. 

Title 

Please avoid numerical numbers on the title; generally, text from titles should be easy to follow. Please correct. 

Author Response

The article entitled "Evaluation of a synthetic retinoid, Ellorarxine (DC645), in the SOD1G93A NSC-34 cell model of ALS" by Escudier O et al., with number ijms-3100888  explores a lead synthetic retinoic acid relevant to ALS, and neuroprotective properties in vitro using the SOD1G93A NSC-34 cell model of ALS interestingly compound displayed pharmacological properties as increased MAP2 expression, increased AMPA receptor GluA2 expression and raised intracellular Ca2+ baseline, as well as antioxidant properties. Although the article is quite interesting, some major concerns should be addressed to publish the article: 

There are a few miscues on redaction and orthography, such as spaces, dots and misswords. Please correct them. 

These have been corrected

Title 

Please avoid numerical numbers on the title; generally, text from titles should be easy to follow. Please correct. 

These have been corrected

 Introduction

Can the chemical structure of Ellorarxine be drawn? This is quite important since it gives an idea of the main changes in the structure used to change activity. 

We have added the chemical structure of Ellorarxine

 Results

Figure images are pretty blurred and don´t distinguish the suggested results. Please try to improve the figure or add a different microphotography; otherwise, the results will be difficult to believe. Please add to the figure legend the number of repetitions you performed in the experiments. 

We agree and have added better quality images, focused on MAP2 and CD200 figures rather than DAPI

Why is DAPI in Figures so blurred and morphology so changed in cells? 

Did you measure the number of cells that successfully induced the expression to have quantitative data rather than qualitative images? If not, why did you do so?

We have shown a representative figure (N=4) showing the clear expression of MAP2 in the cells treated with Ellorarxine (qualitative), together with a quantitative measurement of differentiation (neurite growth) in Figure 2A and B, respectively.

Do you have an image that demonstrates the morphological differentiation on NSC-34 cells? 

We added a figure (Figure 1) showing the differentiation of NSC-34 cells.

What do you mean by Process? Please use the correct terminology for Calcium ions.

We corrected it to “neurite”.Terminology corrected

 Do you evaluate the pharmacokinetic potential of Ellorarxine? 

The PK properties of ellorarxine is not relevant to these in vitro studies. But for information, we have detailed knowledge of the PK of ellorarxine in both rats and mice. The drug is a soluble and very CNS permeable, which gets to target all over the brain, especially areas relevant to MND.

Discussion

Please correct the discussion, which is huge and quite blurred. What are you trying to demonstrate your results? Did your hypothesis get demonstrated? What do you obtain from your results? Why do you believe such results? 

We have shortened the discussion to be a little ore focussed on relevance of results.

The senescence section is difficult to believe since it is not experimentally demonstrated. The authors make hypotheses without any demonstration that are entirely speculative and not quite scientific.

We agree and have removed this speculation

If RNA metabolism is essential, why did you not evaluate it on your assays?

We have clarified this.  

What do you conclude with your experiments? Are you going to improve your compound? Evaluate such on other models? Evaluate toxicity?

We have clarified this in discussion. 

Reviewer 3 Report

Comments and Suggestions for Authors

The authors investigated the mechanistic aspects of Ellorarxine on the SOD1G93A NSC-34 cell model of Amyotrophic Lateral Sclerosis (ALS). The compound increased neurite length, upregulated MAP2 expression, modulated interleukin release, shifted hydrogen peroxide curves rightward, and influenced glutamatergic neurotransmission. The manuscript is well-structured and presents a suitable amount of data. The manuscript is suitable for publication in IJMS with minor revisions.

Specific recommendations include:

  1. Carefully review the use of superscripts and italics throughout the text.
  2. Incorporate a narrative about the discovery or development of Ellorarxine into the introduction.
  3. Include standard deviations for numerical values on lines 107 and 149.
  4. Enhance the descriptions of Figures 4 and 5.

Author Response

This study was performed to probe some key mechanisms relevant to MND in a relevant in vitro model as highlighted by Referee 1, to give rationale for us to take ellorarxine to the clinic (the Three “Ns”, neuroprotection, neuroplasticity, neurorepair”. This has been crucial for us to get rapid MHRA approval.

Comments on the Quality of English Language

The article entitled "Evaluation of a synthetic retinoid, Ellorarxine (DC645), in the SOD1G93A NSC-34 cell model of ALS" by Escudier O et al., with number ijms-3100888  explores a lead synthetic retinoic acid relevant to ALS, and neuroprotective properties in vitro using the SOD1G93A NSC-34 cell model of ALS interestingly compound displayed pharmacological properties as increased MAP2 expression, increased AMPA receptor GluA2 expression and raised intracellular Ca2+ baseline, as well as antioxidant properties. Although the article is quite interesting, some major concerns should be addressed to publish the article: 

There are a few miscues on redaction and orthography, such as spaces, dots and misswords. Please correct them. 

Title 

Please avoid numerical numbers on the title; generally, text from titles should be easy to follow.

We rewrote the title.

The authors investigated the mechanistic aspects of Ellorarxine on the SOD1G93A NSC-34 cell model of Amyotrophic Lateral Sclerosis (ALS). The compound increased neurite length, upregulated MAP2 expression, modulated interleukin release, shifted hydrogen peroxide curves rightward, and influenced glutamatergic neurotransmission. The manuscript is well-structured and presents a suitable amount of data. The manuscript is suitable for publication in IJMS with minor revisions.

We thank the reviewer for the generous words.

Specific recommendations include: 

  1. Carefully review the use of superscripts and italics throughout the text.
  2. Incorporate a narrative about the discovery or development of Ellorarxine into the introduction.
  3. Include standard deviations for numerical values on lines 107 and 149.
  4. Enhance the descriptions of Figures 4 and 5.

These were all corrected.